

# Inter- and intraspecific variation in the *Artibeus* species complex demonstrates size and shape partitioning among species

Brandon P. Hedrick

Cell Biology and Anatomy, Louisiana State University Health Sciences Center- New Orleans, New Orleans, United States of America

## ABSTRACT

Neotropical leaf-nosed bats (family Phyllostomidae) are one of the most diverse mammalian families and *Artibeus* spp. is one of the most speciose phyllostomid genera. In spite of their species diversity, previous work on *Artibeus* crania using linear morphometrics has uncovered limited interspecific variation. This dearth of shape variation suggests that differences in cranial morphology are not contributing to niche partitioning across species, many of which are often found in sympatry. Using two-dimensional geometric morphometric methods on crania from eleven species from the *Artibeus* species complex, the current study demonstrates substantial cranial interspecific variation, sexual size and shape dimorphism, and intraspecific geographic variation. The majority of species were shown to have a unique size and shape, which suggests that each species may be taking advantage of slightly different ecological resources. Further, both sexual size and shape dimorphism were significant in the *Artibeus* species complex. Male and female *Artibeus* are known to have sex specific foraging strategies, with males eating near their roosts and females feeding further from their roosts. The presence of cranial sexual dimorphism in the *Artibeus* species complex, combined with previous work showing that different fruit size and hardness is correlated with different cranial shapes in phyllostomids, indicates that the males and females may be utilizing different food resources, leading to divergent cranial morphotypes. Additional field studies will be required to confirm this emergent hypothesis. Finally, significant geographical shape variation was found in a large intraspecific sample of *Artibeus lituratus* crania. However, this variation was not correlated with latitude and instead may be linked to local environmental factors. Additional work on ecology and behavior in the *Artibeus* species complex underlying the morphological variation uncovered in this study will allow for a better understanding of how the group has reached its present diversity.

# INTRODUCTION

Of the nearly 6500 species of extant mammals, more than 60% are rodents or bats (*Burgin et al., 2018*). Numerous factors such as novel functional innovation (the ever-growing

Corresponding author
Brandon P. Hedrick,
bphedrick1@gmail.com

teeth of rodents, wings of bats) and the repeated evolution of successful locomotor and foraging modes have led these mammal groups to diversify across the globe (*Arita & Fenton, 1997*; *Kay & Hoekstra, 2008*; *Morales et al., 2019*; *Hedrick et al., 2020*). One family of bats, the Neotropical leaf-nosed bats (Family Phyllostomidae) are often considered to represent an adaptive radiation and have diversified into more dietary niches than any other mammalian family (*Freeman, 2000*; *Jones, Bininda-Emonds & Gittleman, 2005*; *Dumont et al., 2012*; *Dumont et al., 2014*; *Shi & Rabosky, 2015*). Ancestrally insectivorous, phyllostomids have expanded into also eating fruit, nectar, blood, and leaves. Members of the Phyllostomidae are widespread throughout the Neotropics and are found in southern North America and throughout Central and South America. In spite of their dietary breadth and expansion across a wide geographical range, all phyllostomids are relatively small bats (7–200 g). How phyllostomids, and bats more generally, have been able to achieve their species diversity,  especially when such a large number of species in the tropics live in sympatry, is an open question.

Artibeus is one of the most speciose genera in the Phyllostomidae (*Lim et al., 2008*) and has been considered to be a model for understanding bat diversification (*Ferreira et al., 2014*). The *Artibeus* species complex is composed of two groups, colloquially termed small *Artibeus* and large *Artibeus*, which are sometimes split at the genus level into *Dermanura* spp. (corresponding to small *Artibeus*) and *Artibeus* spp. (corresponding to large *Artibeus*) (*Redondo et al., 2008*). All members of the *Artibeus* species complex eat fruit, subsisting primarily on figs, and the species complex spans geographically from Mexico to Argentina. The *Artibeus* species complex has also been shown to have relatively limited morphological variation (*Balseiro, Mancina & Guerrero, 2009*; *Marchán-Rivadeneira et al., 2010*). However, size is an important discriminator among species (*Ortega & Castro-Arellano, 2001*; *Larsen, Marchán-Rivadeneira & Baker, 2010*) and may be one of the main factors allowing the *Artibeus* species complex to spread into separate niches. A better understanding of interspecific and intraspecific variation within the *Artibeus* species complex will grant further insight into not only the radiation of the Phyllostomidae, but also the diversification of mammalian species that live in sympatry.

Morphometrics is commonly used to quantify morphological shape for comparison with ecological variables to better understand how and why morphological variation within a group exists (*Samuels & Van Valkenburgh, 2008*; *Moore et al., 2015*; *Vander Linden et al., 2019*; *Hedrick et al., 2019a*; *Hedrick et al., 2019b*).  Caliper-based linear morphometrics have been previously used to assess interspecific variability in *Artibeus* (*Lim, 1997*; *Guerrero, De Luna & Sánchez-Hernández, 2003*; *Lim et al., 2008*; *Balseiro, Mancina & Guerrero, 2009*; *Larsen, Marchán-Rivadeneira & Baker, 2010*; *Marchán-Rivadeneira et al., 2010*), and these studies largely found that there was low interspecific variation across members of the genus. However, geometric morphometrics has been shown to be able to distinguish shape trends more effectively than linear morphometrics (*Mutanen & Pretorius, 2007*; *Zelditch, Swiderski & Sheets, 2012*; *Schmieder et al., 2015*). Thus the relatively small differences in skull shape and size among *Artibeus* species will potentially be easier to capture and distinguish using geometric morphometrics in comparison with linear morphometrics as it captures shape more holistically and allows for better separation of size and shape. This could potentially

uncover previously unknown morphological size and shape partitioning in the *Artibeus* species complex. However, geometric morphometric analyses have yet to be used to analyze variation in *Artibeus*.

The overarching goal of this study is to expand upon previous studies by employing geometric morphometrics on a sample of 279 crania for eleven species of the *Artibeus* species complex (22 species total) to address the following questions: (1) Do different members of the *Artibeus* species complex differ in size and shape? Significant differences between species may suggest differences in niche partitioning via morphological innovation or conversely suggest non-adaptive processes such as genetic drift or non-morphological adaptive bases for niche partitioning (e.g., echolocation differences, biting behavior). (2) How have factors such as phylogeny, sexual dimorphism, and geography structured that variation? Interspecific analyses are done to evaluate differences in size, shape, and sexual dimorphism across the *Artibeus* species complex. Based on previous studies (*Marchán-Rivadeneira et al., 2010*), I predict that the *Artibeus* species complex will separate into two clear size groups (small and large *Artibeus*) and that there will be additional separation within those two groups by size. Similarly, I predict that shape will not differ substantially across species following previous studies based on linear morphometrics (*Balseiro, Mancina & Guerrero, 2009*; *Marchán-Rivadeneira et al., 2010*). I predict significant sexual dimorphism based on previous studies that have found male and female *Artibeus* species to exhibit differing foraging behaviors (*Kunz & Diaz, 1995*). To assess intraspecific variation and geographical variation in the *Artibeus* species complex, *A. lituratus* is heavily sampled across its range. I hypothesize that there is shape and size variation across *A. lituratus'* range based on previous studies finding correlations between *A. lituratus* skull shape and environmental variables (*Marchan-Rivadeneira et al., 2012*). Using these data, I address the degree of morphological variation in the *Artibeus* species complex, discuss the underlying causes for that variation, and shed light on the radiation of the *Artibeus* complex.

## MATERIALS & METHODS

### Materials and initial analyses

*Artibeus* spp. ($n = 279$) crania were photographed in lateral and ventral view at the Louisiana State University Museum of Natural Sciences (LSUMZ) using a Canon EOS 70D fitted with a Canon EF-S 60 mm f/2.8 Macro USM fixed lens mounted on a photostand to ensure the same angle was used in each picture. Two separate datasets were used, one to analyze interspecific variation ($n = 186$) and one to analyze intraspecific variation in *A. lituratus* ($n = 113$). The interspecific sample included *A. anderseni* ($n = 8$), *A. phaeotis* ($n = 20$), *A. cinereus* ($n = 20$), *A. toltecus* ($n = 14$), *A. aztecus* ($n = 9$), *A. fraterculus* ($n = 20$), *A. obscurus* ($n = 20$), *A. jamaicensis* ($n = 20$), *A. planirostris* ($n = 20$), *A. lituratus* ($n = 20$), and *A. fimbriatus* ($n = 15$) (Fig. 1). Classifications were done according to *Rojas, Warsi & Dávalos (2016)* who considered all of these species to belong to *Artibeus*, but note that *A. anderseni, A. aztecus, A. cinereus, A. phaeotis*, and *A. toltecus* are also sometimes classified as *Dermanura*. To reduce confusion, the term '*Artibeus* species complex' is used in this study to refer to all species included in this study.

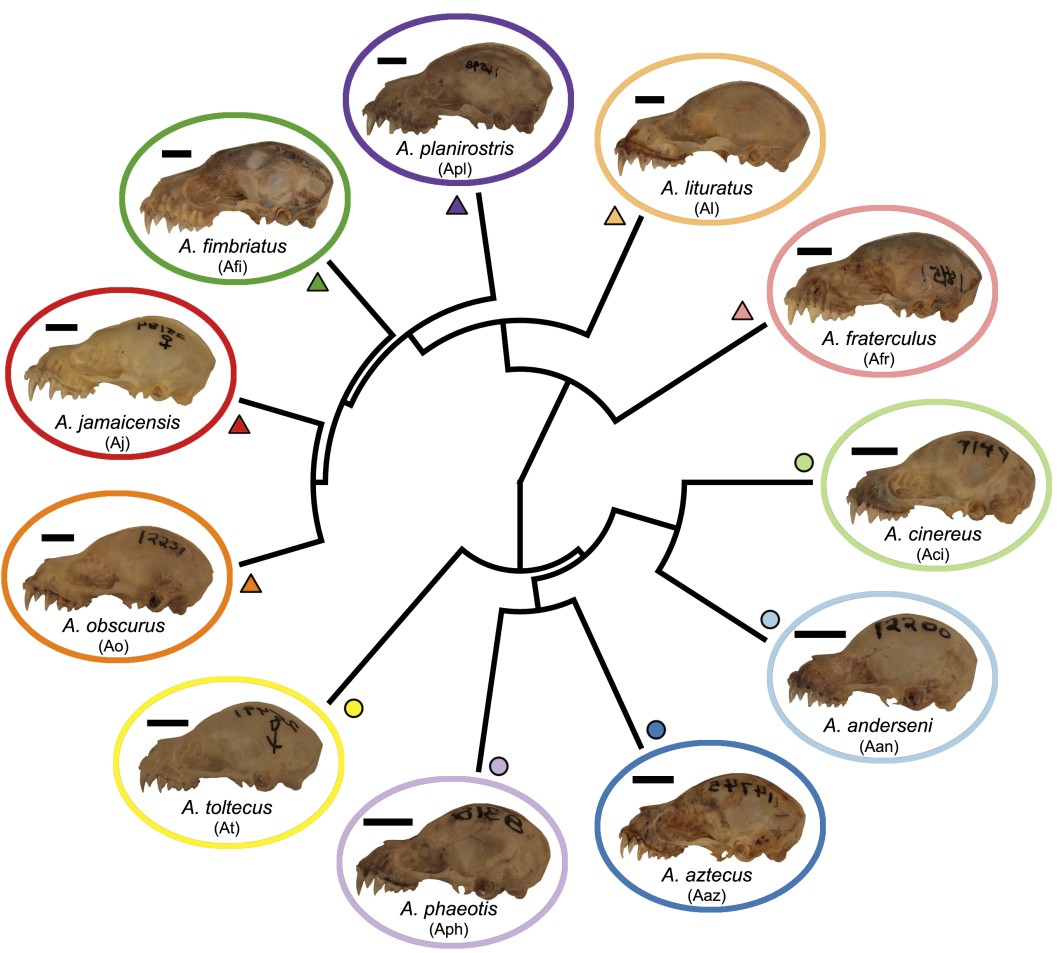

**Figure 1** **Phylogeny of the *Artibeus.* species complex.** Representative skulls shown in lateral view with scale bars. Scale = 50 mm. Each skull has its respective species code used in Figs. 3 and 4 (e.g., Aj = *Artibeus jamaicensis*). Colored shapes correspond to species in Figs. 3 and 4. Large *Artibeus* are represented by triangles and small *Artibeus* are represented by circles. The phylogeny was generated based on *Rojas, Warsi & Dávalos (2016)*.

The intraspecific sample included populations of *A. lituratus* from Argentina ($n = 20$), Belize ($n = 6$), Costa Rica ($n = 11$), Paraguay ($n = 20$), Peru ($n = 20$), Trinidad ($n = 7$), and two localities in Mexico (Colima, $n = 8$; Tabasco, $n = 21$). Each population here is named based on the country of origin. 'Countries' were used as proxies for populations due to limited specimen availability at individual localities in the visited museum collections. Generally, all specimens from each 'country' are quite close to one another geographically. Two populations were sampled in Mexico, which are ∼1,500 km apart and are thus referred by the Mexican state where they were collected. Full details regarding collection information are present in the Supplemental Information. *A. lituratus* was selected for analyzing intraspecific variation in *Artibeus* based on its large range and its abundance and availability in the LSUMZ collections. I aimed to include 20 specimens for each species and country respectively, but that was not always possible due to limitations in the museum

collections. All specimens were coded for sex and I aimed to have equal numbers of males and females in each sample (53 males, 60 females; see Supplemental Information). To keep sample sizes balanced in the interspecific analyses, the Argentina *A. lituratus* specimens ($n = 20$) were included in the interspecific dataset. *A. lituratus* specimens from other localities were only examined in the intraspecific dataset.

Skulls were landmarked and semi-landmarked in tpsDIG2 (*Rohlf, 2006*) in lateral and ventral views. Landmarks represent homologous anatomical loci while semi-landmarks represent homologous curves. Eleven landmarks and one semi-landmark curve consisting of 15 semi-landmarks were digitized in lateral view and seventeen landmarks and one semi-landmark curve consisting of 10 semi-landmarks were digitized in ventral view (Fig. 2, Table S1). Landmarks were imported into RStudio v. 1.1.463 (*R Core Development Team, 2019*) and opened in *geomorph* v. 3.0.7 (*Adams & Otárola-Castillo, 2013*). Landmarks were subjected to Generalized Procrustes Analysis (GPA) and semi-landmarks were evenly spaced and were slid according to the bending energy criterion (*Bookstein, 1997*; *Perez, Bernal & Gonzalez, 2006*; *Zelditch, Swiderski & Sheets, 2012*). This generated two centroid sizes per specimen (one lateral view and one ventral view). Each centroid size was used in analyses with its corresponding shape data (e.g., lateral view centroid size analyzed with lateral view shape data). An error analysis was performed whereby one individual (LSUMZ 9425) was landmarked 10 additional times in both lateral and ventral views to assess within individual landmarking error. All replicates of this specimen were then plotted in principal component morphospace to evaluate general shape trends and ensure that the replicates of the same individual clustered together closely with other replicates (Fig. S1), prior to running analyses on the dataset as a whole.

### Interspecific data analyses

To analyze trends in the data, both lateral and ventral landmark configurations were subjected to principal component analysis (PCA). Principal components (PCs) that represented greater than 10% of total shape variation were examined. To address whether shape varied by size for both the lateral and ventral datasets, relationships between the common allometric component (CAC) of shape variation (Mitteroecker et al., 2004) were tested against log10-transformed centroid size using a Procrustes ANOVA that was permuted 999 times. The CAC represents the allometric trend in the data. These data were represented graphically both as regressions and using violin plots of log10-transformed centroid size. Additionally, Procrustes ANOVAs were done to analyze the relationship between shape, species, sex, and size (shape ~species * sex * size) in both views where size was represented by log10-transformed centroid size. Finally, sexual size dimorphism was analyzed in both views (size ~species * sex). Due to sample size limitations, sexual dimorphism was examined in the *Artibeus* species complex as a whole, and in *A. lituratus*, which was sampled at a higher rate for the intraspecific analyses (see below).

Species means of shape for lateral and ventral views were then taken for all 11 species to quantify the degree of phylogenetic signal in the data. Species means were first subjected to GPA. Shape data and log10-transformed centroid sizes were evaluated for phylogenetic signal using Blomberg's K statistic modified for multivariate shape data (Kmult) (*Blomberg,*

 

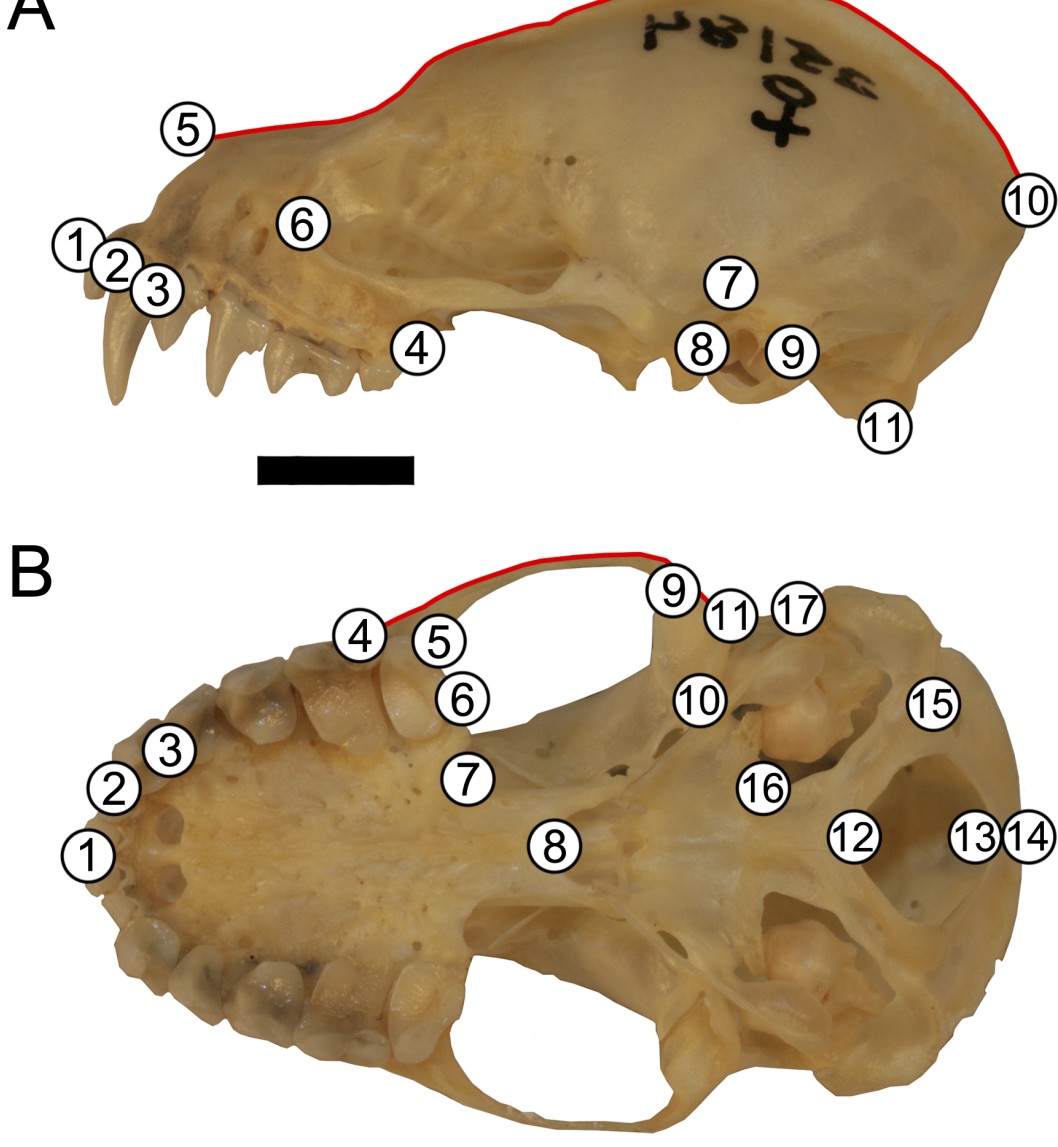

**Figure 2  Landmark configuration for specimens in (A) lateral and (B) ventral views.** Numbered circles correspond to landmarks while red lines correspond to semi-landmark curves. See Table S1 for landmark definitions. Scale = 50 mm.

*Garland & Ives, 2003*; *Adams, 2014*) in *geomorph*. This was done  using the phylogeny from *Rojas, Warsi & Dávalos (2016)*. In addition to the Procrustes ANOVAs discussed above, phylogenetically-corrected Procrustes ANOVAs were performed on means data. This required combining each species into a single mean (thereby losing the sex component). The species phylogeny was plotted onto a PCA of mean shapes to visually examine how phylogeny and morphology related to one another in morphospace. To better understand the evolutionary processes that underlie phylogenetic signal in the data, patterns of disparity accumulation through time were calculated using the dtt function in *geiger* (*Harmon et*

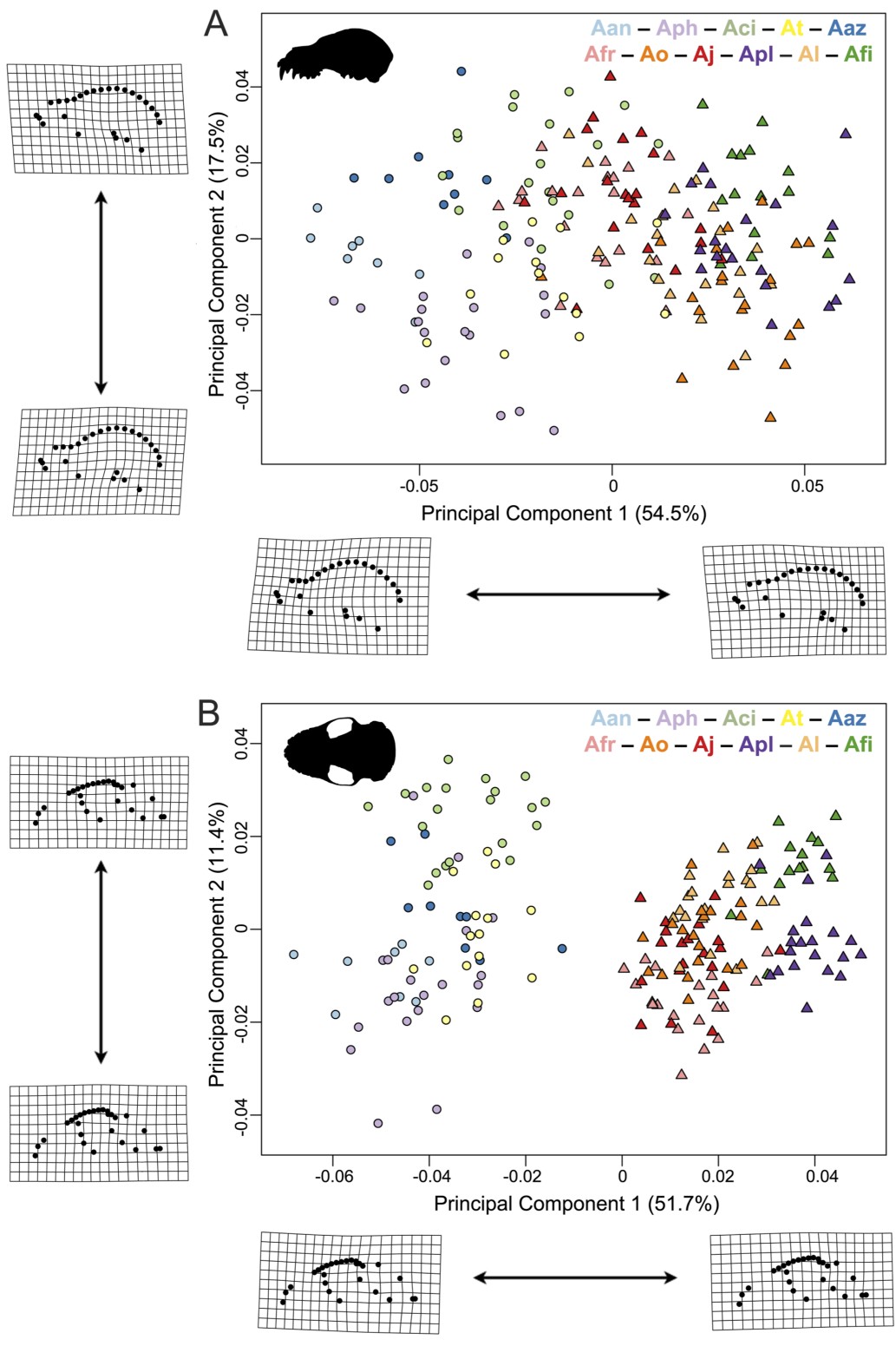

**Figure 3** Principal component analyses of the *Artibeus* species complex in (A) lateral view and (B) ventral view. Thin-plate spline grids represent shape change along principal component 1 and 2. Colors and species codes refer to individual taxa within the *Artibeus* species complex and are derived from Fig. 1.

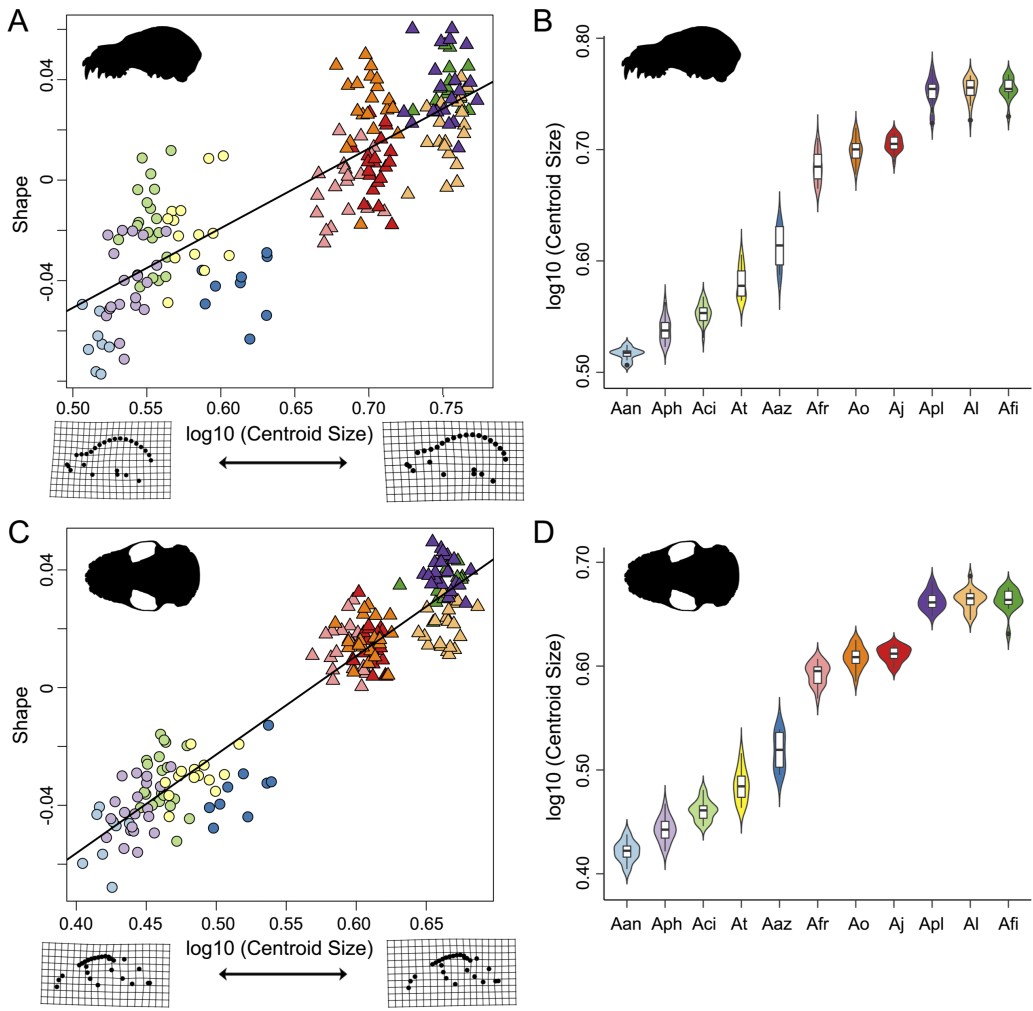

**Figure 4** **Evolutionary allometric trajectories for specimens of the *Artibeus* species complex in (A) lateral view and (C) ventral view.** These plots show the common allometric component (CAC) of shape (*y*-axis) plotted against the log10-transformed centroid size (*x*-axis). Centroid size box plots for (B) lateral view and (D) ventral view. Colors and species codes refer to individual taxa within the *Artibeus* species complex and are derived from Fig. 1.

*al., 2008*). By analyzing disparity through time, it is possible to better understand whether shape evolved under a Brownian motion pattern or through punctuated bursts of disparity.

In order to further analyze differences between species, the *nicheROVER* package (*Swanson et al., 2015*) was utilized. This analysis was developed to evaluate niche overlap with multidimensional data and has been previously applied to shape data (*Machado, Zahn & Marroig, 2018*). Separate analyses were run for the lateral and ventral datasets utilizing principal components 1 and 2 and log10-transformed centroid size to quantitatively assess how much overlap is present in shape and size between species pairs. Posterior distributions were generated for these values for each species using 10,000 permutations. The overlap function was then used to calculate overlap metric estimates at a 95% confidence interval.

Finally, the posterior probabilities that a given species overlaps with another species in size and shape were plotted.

## Intraspecific data analyses

Lateral and ventral *A. lituratus* crania were subjected to PCA as was done in the interspecific analyses. Procrustes ANOVAs were performed to assess the relationship between shape, sex, size, and country (shape ~sex + size + country) in both views, with size represented by log10-transformed centroid size. Posthoc tests were performed using the advanced.procD.lm function in *geomorph* to generate pairwise comparisons (*Collyer, Sekora & Adams, 2015*). Sexual size dimorphism was assessed in base R (size ~country + sex) for both views.

To visualize how close individuals in portions of the geographic range (countries) cluster to one another, distance matrices based on Euclidean distances were calculated for country means in base R using all principal components for both lateral and ventral datasets (*R Core Development Team, 2019*). Hierarchical clustering was then performed using unweighted pair-group method with arithmetic means (UPGMA) through the hclust function. These clusters were then plotted as dendrograms. To determine how well the dendrograms reflect the underlying hierarchical structure, the cophenetic correlation coefficient was calculated (*Sneath & Sokal, 1973*). This compares heights of tips above the node that they are joined to with the observed correlation matrix (*Zelditch, Swiderski & Sheets, 2012*).

## RESULTS

### Interspecific results

In principal component space, large *Artibeus* and small *Artibeus* groups separated in the lateral and ventral datasets along PC1 (Fig. 3, Fig. S2 for colorblind version). The separation is less clear in the lateral dataset than the ventral dataset (see Table S2 for full results).

In the lateral PCA, PC1 accounted for 54.5% of total variation. The small *Artibeus* (*A. anderseni*, *A. phaeotis*, *A. cinereus*, *A. toltecus*, and *A. aztecus*) plotted on the negative PC1 axis while the large *Artibeus* (*A. fraterculus*, *A. obscurus*, *A. jamaicensis*, *A. planirostris*, *A. lituratus*, and *A. fimbriatus*) plotted on the positive PC1 axis. While PC1 was largely structured by size, some of the largest (e.g., *A. lituratus*) and smallest (e.g., *A. cinereus*) species plotted towards the center of morphospace rather than on the ends defined by PC1. Therefore, PC1 incorporated both size and shape data. Taxa on the negative end of PC1 tended to have a shorter rostrum than those on the positive end of PC1 (Fig. 3A). PC2 accounted for 17.5% of total variation and separated out species within larger clusters. Specimens on the positive end of PC2 had a less pronounced dome shape to their crania than those on the negative end of PC2. Specimens on the negative end of PC2 also had a more caudally positioned zygomatic arch relative to the position of the external acoustic meatus. Within the small *Artibeus* cluster, *A. phaeotis*, *A. toltecus*, and *A. anderseni* plotted on the negative PC2 axis while *A. aztecus* and *A. cinereus* plotted on the positive PC2 axis. *A. cinereus* overlapped strongly with *A. jamaicensis* and *A. fraterculus*. In the large *Artibeus* cluster, all taxa plotted in one large poorly differentiated group. However, *A. fimbriatus*

and *A. planirostris* tended to have higher PC1 scores than the other large *Artibeus* species. Additional PCs accounted for less than 10% of total variance.

Shape and log-transformed centroid size were significantly correlated for the lateral dataset ($p < 0.001$, $R^2 = 0.369$). The small *Artibeus* group and large *Artibeus* group separated in centroid size and shape (Fig. 4A). All small *Artibeus* species occupy different sizes with minimal overlap (Fig. 4B). *A. cinereus* and *A. toltecus* overlap along the common allometric component of shape, but have different size ranges (*A. cinereus* mean CS = 3.57, *A. toltecus* mean CS = 3.81). *A. aztecus* and *A. phaeotis* show a similar situation where they overlap in shape, but not in size (Figs. 4A, 4B). *A. anderseni* is the smallest taxon included (mean CS = 3.28). There are two size groups for the large *Artibeus* taxa, one including *A. fraterculus*, *A. obscurus*, and *A. jamaicensis* and the other including *A. lituratus*, *A. planirostris*, and *A. fimbriatus* (Fig. 4B). Within the smaller large *Artibeus* cluster, *A. fraterculus* and *A. jamaicensis* overlap in shape, but *A. obscurus* does not. In the larger large *Artibeus* cluster, *A. planirostris* and *A. fimbriatus* overlap in shape, but *A. lituratus* does not.

Similar to the lateral PCA, in the ventral dataset large and small *Artibeus* are separated with small *Artibeus* plotting on the negative end of PC1 and large *Artibeus* plotting on the positive end of PC1 (Fig. 3B). Unlike the lateral PCA, the two clusters do not have any overlap. PC1 accounted for 51.7% of the total variance and PC2 accounted for 11.4% of total variance. Additional PCs accounted for less than 10% of the total variance. Taxa on the negative end of PC1 had a relatively shorter rostrum and longer basicranium than those on the positive end of PC1. Along PC1, the large *Artibeus* cluster had a clear separation between an overlapping group composed of *A. lituratus*, *A. jamaicensis*, *A. fraterculus*, and *A. obscurus* and a group composed of *A. fimbriatus* and *A. planirostris*. *A. fimbriatus* and *A. planirostris* were further separated along PC2. Within the small *Artibeus* group, PC2 separated *A. cinereus* from the other small *Artibeus* taxa (*A. phaeotis*, *A. aztecus*, *A. toltecus*, and *A. anderseni*), which overlapped. Specimens on the negative end of PC2 had a wider skull with a more curved zygomatic arch than those on the positive end of PC2.

Size and shape were similarly correlated in the ventral dataset ($p < 0.001$; $R^2 = 0.461$). Compared to the lateral dataset, the taxa had less spread along the common allometric coefficient shape axis (Fig. 4C). The log-transformed centroid size spread for the lateral dataset and the ventral dataset strongly agreed (Figs. 4B, 4D). As in the lateral dataset, the small and large *Artibeus* groups were strongly separated in centroid size. *A. anderseni* differed in size and shape from other small *Artibeus* taxa. However, *A. phaeotis*, *A. cinereus*, *A. toltecus*, and *A. aztecus* all overlapped on the shape axis. The large *Artibeus* taxa split into two size-based groups as in the lateral dataset. Unlike the lateral dataset, *A. fraterculus*, *A. obscurus*, and *A. jamaicensis* overlapped on the shape axis. Similar to the lateral dataset, *A. planirostris*, *A. lituratus*, and *A. fimbriatus* form a cluster. *A. planirostris* and *A. fimbriatus* have similar sizes and shapes whereas *A. lituratus* has a different shape from the other two taxa.

NicheROVER analyses similarly demonstrated minimal overlap between species when size and shape (PC1, 2) axes were considered (Figs. S3, S4). In lateral view, only *A. jamaicensis* and *A. fraterculus* and *A. lituratus*, *A. fimbriatus*, and *A. planirostris* had more

**Table 1  Lateral view Procrustes ANOVA assessing the relationship between shape, species, sex, and log10-transformed centroid size (Shape ~ Species * Sex * Size).** Bolded values represent significant *p*-values at $\alpha = 0.05$.

|  | Df | SS | MS | R$^2$ | F | Z | p |
|---|---|---|---|---|---|---|---|
| Species | 10 | 0.228 | 0.023 | 0.605 | 28.058 | 12.092 | **0.001** |
| Sex | 1 | 0.003 | 0.003 | 0.009 | 4.184 | 4.654 | **0.001** |
| Size | 1 | 0.004 | 0.004 | 0.010 | 4.521 | 5.040 | **0.001** |
| Species:Sex | 10 | 0.010 | 0.001 | 0.026 | 1.183 | 6.596 | **0.001** |
| Species:Size | 10 | 0.010 | 0.001 | 0.026 | 1.197 | 6.733 | **0.001** |
| Sex:Size | 1 | 0.000 | 0.000 | 0.000 | 0.133 | −1.405 | 0.919 |
| Species:Sex:Size | 10 | 0.007 | 0.001 | 0.018 | 0.841 | 5.441 | **0.001** |
| Residuals | 142 | 0.115 | 0.001 | 0.306 | | | |
| Total | 185 | 0.377 | | | | | |

Notes.
   Df, degrees of freedom;  SS, sum of squares;  MS, mean squares..

**Table 2  Ventral view Procrustes ANOVA assessing the relationship between shape, species, sex, and log10-transformed centroid size (Shape ~ Species * Sex * Size).** Bolded values represent significant *p*-values at $\alpha = 0.05$.

|  | Df | SS | MS | R$^2$ | F | Z | p |
|---|---|---|---|---|---|---|---|
| Species | 10 | 0.220 | 0.022 | 0.630 | 31.324 | 13.058 | **0.001** |
| Sex | 1 | 0.001 | 0.001 | 0.002 | 1.149 | 3.081 | **0.001** |
| Size | 1 | 0.003 | 0.003 | 0.010 | 4.922 | 6.701 | **0.001** |
| Species:Sex | 10 | 0.008 | 0.001 | 0.023 | 1.151 | 9.060 | **0.001** |
| Species:Size | 10 | 0.010 | 0.001 | 0.027 | 1.366 | 10.198 | **0.001** |
| Sex:Size | 1 | 0.001 | 0.001 | 0.002 | 0.915 | 3.091 | **0.003** |
| Species:Sex:Size | 10 | 0.007 | 0.001 | 0.020 | 0.970 | 8.133 | **0.001** |
| Residuals | 142 | 0.100 | 0.001 | 0.286 | | | |
| Total | 185 | 0.350 | | | | | |

Notes.
   Df, degrees of freedom;  SS, sum of squares;  MS, mean squares..

than 50% mean overlap. In ventral view, *A. jamaicensis*, *A. fimbriatus*, and *A. obscurus* had over 50% mean overlap, but other species did not.

Based on the Procrustes ANOVA of the lateral shape dataset (Table 1, Table S3), all main factors (species, sex, and size) were significant ($p < 0.001$). Further, the interaction of species and sex was also significant ($p < 0.001$) showing that different sexes are different shapes for different species (intraspecific sexual shape dimorphism). The interaction of species and size was also significant ($p < 0.001$). Shape and the interaction of sex and size was not significant ($p = 0.919$). Finally, the interaction of species, sex, and size was significant with respect to shape ($p < 0.001$) showing that allometric relations change with species and with each sex in each species. The trends for the ventral dataset (Table 2) were the same except that the interaction of sex and size was significant ($p = 0.003$).

In lateral view, both shape ($K_{mult} = 0.80$, $p = 0.008$) and size ($K_{mult} = 2.51$, $p < 0.001$) were significantly correlated with phylogeny. Similarly, the ventral dataset showed that both shape ($K_{mult} = 1.04$, $p = 0.007$) and size ($K_{mult} = 2.50$, $p < 0.001$) were significantly

correlated with phylogeny (see Table S4 for PCA results). Mapping the phylogeny on the species means in principal component space demonstrates a separation between small and large *Artibeus* in morphospace with less clear trends within each group (Figs. S5A, S5B). In lateral view, there is a peak in disparity at approximately 5 mya, which is at the timing of the split between small and large *Artibeus* (Fig. S5C). This suggests a burst in morphological disparity at that time. Disparity accumulation in ventral view generally follows Brownian motion with a small increase in disparity at 5 mya (Fig. S5D). Using phylogenetically corrected Procrustes ANOVAs, neither size nor species were significantly correlated with lateral or ventral shape (Table S4). This is likely due to the strong separation in both size and shape between the small *Artibeus* and large *Artibeus*, which each form a monophyletic group. Given a large amount of individual variation and species overlap, the mean shapes are likely simplifying the trends and reducing the signal. Therefore, the discussion will largely concern the non-phylogenetically corrected results recognizing that there is a phylogenetic component to cranial shape in the *Artibeus* species complex.

### Intraspecific results

Although there was a substantial amount of overlap in principal component space, *A. lituratus* specimens grouped by locality (country) in both the lateral and ventral datasets, demonstrating some intraspecific geographic variation within *A. lituratus* (Fig. 5; see Table S5 for full results).

In the lateral dataset, PC1 accounted for 31.7% of the total variation and PC2 accounted for 24.1% of the total variation (Fig. 6A). Specimens on the positive PC1 axis had relatively longer zygomatic arches and a somewhat more compact skull. Specimens from Argentina exemplified this morphotype. Specimens on the negative end of PC1 had shorter zygomatic arches with a less compact skull, shown by specimens from Trinidad. While these specimens group by country in morphospace, it should be noted that the shape differences are quite minor. In the ventral dataset, PC1 accounted for 25.9% of total variation while PC2 accounted for 13.4% of total variation (Fig. 6C). Specimens on the positive PC1 axis had somewhat more compact skulls with flatter zygomatic aches, exemplified by specimens from Argentina and Paraguay, than taxa at the negative end of PC1. However, as with the lateral dataset, these shape differences were quite minor in spite of clear country-based groupings in morphospace.

Based on a Procrustes ANOVA (lateral shape ~sex + size + country), lateral shape was significantly correlated with sex ($p = 0.002$), size ($p < 0.001$), and country ($p < 0.001$). Additional pairwise tests were performed to establish significances between countries using the advanced.procD.lm function in *geomorph* (See Table 3, above diagonal for lateral view; for Z-scores, see Table S6). Cranial centroid size in lateral view was lower in the northern localities (Belize, Mexico, Costa Rica) in comparison with the southern ones (Peru, Trinidad, Paraguay, Argentina) (Table 4). The ventral shape dataset was significantly corelated with size ($p < 0.001$) and country ($p < 0.001$), but not sex ($p = 0.39$) under the Procrustes ANOVA model (ventral shape ~sex + size + country). Pairwise comparisons showed fewer significant differences in the ventral dataset in comparison with the lateral dataset (Table 3, below diagonal for ventral view; for Z-scores, see Table S6). Size was

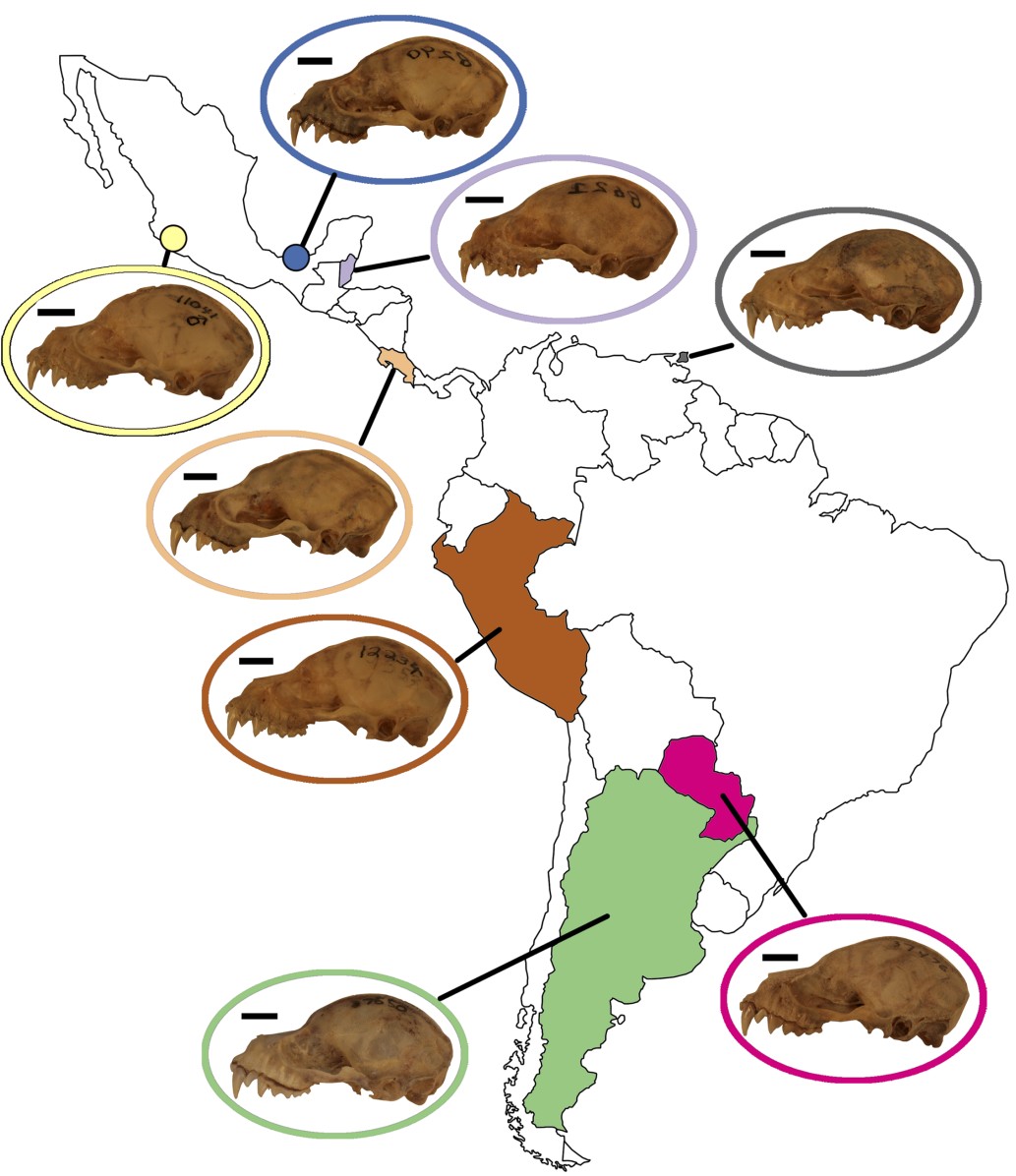

**Figure 5** **Map of Mexico, Central, and South America highlighting countries where samples of *A. lituratus* were taken.** Representative skulls shown in lateral view with scale bars. Scale = 50 mm. Colored circles around skulls and countries correspond to colors used to identify countries in Fig. 6.

significantly correlated with country in both lateral ($p < 0.001$) and ventral ($p < 0.001$) views. Sex was correlated with size in lateral view ($p = 0.027$), but was marginally significant in ventral view ($p = 0.063$). The average centroid size in ventral view strongly correlated with the average centroid size in lateral view such that the more northern countries had smaller average centroid sizes than the more southern countries (Table 4).

The UPGMA analyses had cophenetic coefficients that suggest that the dendrograms do display underlying hierarchy (lateral dataset cophenetic coefficient = 0.85; ventral dataset

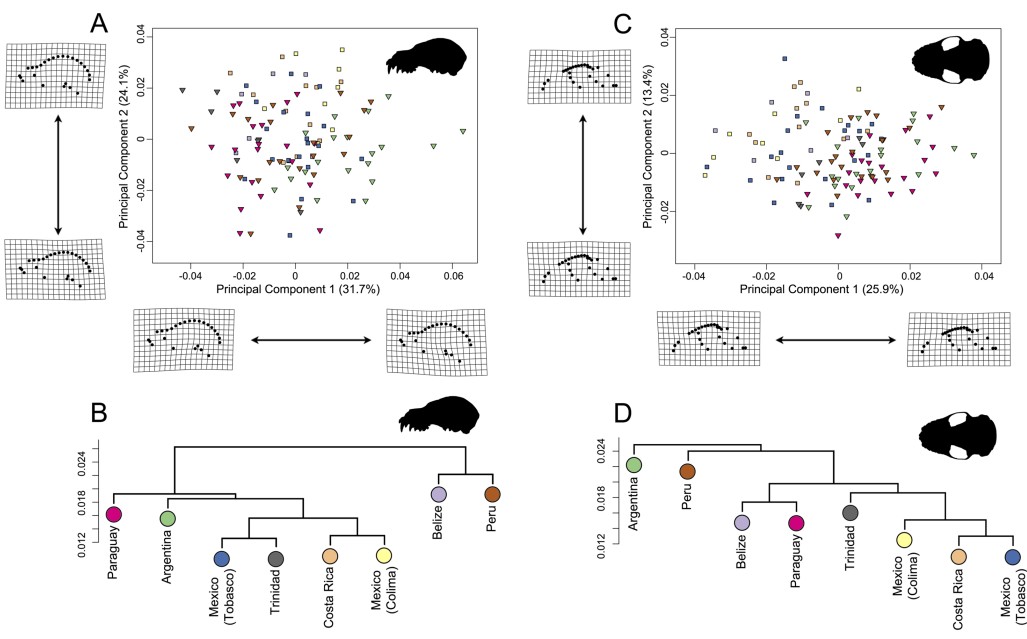

**Figure 6** **Principal component analyses for *A. lituratus* subset in (A) lateral view and (C) ventral view. UPGMA cluster plots for (B) lateral view and (D) ventral view.** Colors refer to countries where *A. lituratus* specimens were collected and are derived from Fig. 5. Southern localities (Argentina, Paraguay, Peru, Trinidad) are inverted triangles while northern localities (Mexico, Belize, Costa Rica) are squares.

**Table 3** **Pairwise comparison *p*-values between countries for the *A. lituratus* data (lateral view above diagonal, ventral view below diagonal).** (Shape ∼ Sex + Size + Country).

|     | Arg   | Bel   | Cos   | Col   | Tab   | Par   | Per   | Tri   |
|-----|-------|-------|-------|-------|-------|-------|-------|-------|
| Arg |       | **0.001** | **0.001** | **0.001** | **0.001** | **0.001** | **0.001** | **0.001** |
| Bel | **0.001** |       | 0.257 | **0.044** | **0.039** | 0.064 | **0.035** | 0.535 |
| Cos | **0.001** | 0.283 |       | 0.569 | 0.079 | **0.006** | **0.033** | **0.037** |
| Col | **0.001** | 0.791 | 0.164 |       | **0.013** | **0.002** | **0.013** | **0.011** |
| Tab | **0.001** | 0.157 | 0.269 | 0.084 |       | 0.116 | 0.595 | **0.037** |
| Par | 0.096 | **0.001** | **0.001** | **0.001** | **0.001** |       | **0.041** | 0.189 |
| Per | 0.088 | **0.006** | **0.002** | **0.001** | **0.019** | **0.004** |       | 0.016 |
| Tri | 0.19  | **0.045** | 0.108 | **0.014** | 0.417 | **0.035** | 0.472 |       |

**Notes.**

Arg, Argentina; Bel, Belize; Cos, Costa Rica; Col, Mexico Colima; Tab, Mexico, Tabasco; Par, Paraguay; Per, Peru; Tri, Trinidad.

Bolded values represent significant *p*-values at α = 0.05. *Z*-scores are presented in Table S6 for both lateral and ventral views.

cophenetic coefficient = 0.83). In both datasets, the Mexican localities, Costa Rica, and Trinidad formed a cluster (Figs. 6B; 6D). This did not include Belize in either case. In the lateral dataset, Belize and Peru grouped together and Argentina and Paraguay grouped together. In ventral view, Belize and Paraguay grouped together and Argentina and Peru grouped together.

**Table 4 Average centroid sizes for *A. lituratus* specimens by country with smallest values at the top and largest at the bottom.** The order for the lateral view and ventral view is identical.

|  | Average centroid size lateral view | Average centroid size ventral view |
|---|---|---|
| Belize | 5.27 | 4.24 |
| Mexico, Colima | 5.33 | 4.33 |
| Costa Rica | 5.54 | 4.48 |
| Mexico, Tabasco | 5.55 | 4.53 |
| Argentina | 5.68 | 4.61 |
| Trinidad | 5.72 | 4.64 |
| Peru | 5.73 | 4.65 |
| Paraguay | 5.78 | 4.69 |

# DISCUSSION

Speciation can occur through numerous avenues, including vicariance or via modifications in resource use of species in sympatry. Quantifying morphological variation can grant insight into how species may be converging on similar morphologies or diverging into disparate morphologies, whether that is occurring interspecifically or intraspecifically across a wide geographical range. As a result, such analyses help to shed light onto the underlying mechanisms structuring the degree of morphological variation in a group. Bats are one of the most diverse mammalian groups, with numerous species living in sympatry with closely related species. The phyllostomids are one of the most diverse bat families (*Shi & Rabosky, 2015*) and the *Artibeus* species complex is highly diverse within the Phyllostomidae, making them an excellent group with which to ask questions about interspecific and intraspecific morphological variation. The present study demonstrates that species within the *Artibeus* species complex have diverged into discordant morphologies and/or sizes, potentially allowing them to occupy different niches. Further, significant intraspecific differences were found within *A. lituratus* across its range, demonstrating that within species variation is an important factor to consider when analyzing skull shape in bats. However, no clear patterns of geographical variation were found.

## Interspecific variation of the *Artibeus* species complex

This study demonstrated clear species level differences in cranial size and shape across the *Artibeus* species complex (Tables 1 and 2), which may suggest niche partitioning of species if these differences impact biting performance or resource use. The present study confirmed the broad size difference between small and large *Artibeus* found in prior studies that used linear morphometrics (*Lim, 1997*; *Ortega & Castro-Arellano, 2001*; *Larsen, Marchán-Rivadeneira & Baker, 2010*; *Marchán-Rivadeneira et al., 2010*). Principal component 1 in lateral and ventral views were largely related to size and showed clear separation between these two groups (Fig. 3). This result was corroborated by an allometric analysis demonstrating a significant relationship between size and shape with small *Artibeus* species separating fully from large *Artibeus* species in both lateral (Figs. 4A) and ventral

(4C) views. It was further shown by a lack of overlap in size and shape variables (Figs. S3, S4).

In addition to the division between the broader small and large *Artibeus* groups, the present analysis found fine-scale size and shape-based divisions within each group. While some previous studies have found species level differences in cranial shape variation in the *Artibeus* species complex (*Lim et al., 2008*), the majority of previous studies have found that cranial shape does not substantially vary within either large or small *Artibeus* groups (e.g., *Balseiro, Mancina & Guerrero, 2009*; *Marchán-Rivadeneira et al., 2010*). Two clear size-based groups were found in the large *Artibeus*, which are present in both lateral view (Fig. 4B) and ventral view (Fig. 4D). The first group included *A. fraterculus*, *A. obscurus*, and *A. jamaicensis* and the second included *A. planirostris*, *A. lituratus*, and *A. fimbriatus*. This is in agreement with *Ortega & Castro-Arellano (2001)*, who noted that *A. planirostris* and *A. fraterculus* differed in size. Variation in shape across these species was also present, with *A. obscurus* having a different shape from *A. fraterculus* and *A. jamaicensis* in the first group and *A. lituratus* having a different shape than *A. planirostris* and *A. fimbriatus* in the second group (Figs. 4A, 4C). Additionally, the small *Artibeus* had clear size separation across all species examined (*A. anderseni*, *A. phaeotis*, *A. cinereus*, *A. toltecus*, and *A. aztecus*). Just like with the large *Artibeus*, the small *Artibeus* differed in shape as well, with *A. cinereus* and *A. toltecus* having similar shapes, which were different from *A. phaeotis* and *A. aztecus*, especially in lateral view (Fig. 4A). *A. anderseni* had a different shape from all small *Artibeus* species. Overall, the majority of species differed from one another in cranial size, shape, or a combination of the two (Fig. S3; S4).

Phylogeny was significantly correlated with skull shape in both the lateral and ventral datasets. Phylogenetic comparative methods using species means showed a lack of significance between shape, size, and species (Table S4), which may suggest that some of the shape differences across species uncovered in the non-phylogenetically corrected ANOVAs may be related to allometric differences or drift rather than morphological changes related to skull function. This lack of significance may also be related to the strong divide in both shape and size between large and small *Artibeus*. Examining shape data of species means in a phylomorphospace revealed strong clustering of small *Artibeus* and large *Artibeus* species, but less phylogenetic influence within each broader group (Figs. S5A, S5B). Evaluating disparity through time demonstrated a spike in disparity at the time when the small and large *Artibeus* split (Fig. S5C, S5D). This supports the idea that the main phylogenetic influence in the data was the division between small and large *Artibeus*, which are sometimes considered separate genera (*Redondo et al., 2008*).

Behavioral plasticity in feeding habits can drive variation both within species and between species. *Dumont & O'Neal (2004)* found that different pteropodid species perform different biting behaviors when eating fruits of different hardnesses and suggested that some species access hard fruit diets by changing behavior while others may do so through morphological changes. Recently, *Hedrick et al. (2020)* showed that it is likely behavioral plasticity rather than morphological shape change of their proximal limb bones that has allowed diversification of rodents into varied locomotor niches, suggesting that behavioral plasticity rather than morphological change can facilitate diversification. Therefore, it is

likely that morphological variation is not the only factor that has allowed the *Artibeus* species complex to proliferate, and that behavioral plasticity may be another factor. Although figs make up 78% of the diet of *Artibeus jamaicensis* (*Ortega & Castro-Arellano, 2001*) and *Artibeus* spp. has been considered a specialist on Cecropiaceae and Moraceae (*Fleming, 1986*), fig trees are not available as a food resource at all times of the year (*August, 1981*). For example, *A. jamaicensis* and *A. lituratus* have been shown to be folivorous for part of the year (*Zortea & Mendes, 1993*; (*Kunz & Diaz, 1995*) and *A. lituratus* is known to have a more generalist diet in regions where figs are not as abundant (*Galetti & Morellato, 1994*). Given annual dietary variation, a combination of morphological and behavioral plasticity may have led *Artibeus* spp. to have a previously unrecognized amount of both interspecific and intraspecific variation. Small *Artibeus* species tend to have a taller, more dome shaped skull with a shorter rostrum, indicating a higher mechanical advantage than the less tall, rostrocaudally long skulls of large *Artibeus* species (Fig. 3). Since the small *Artibeus* species have smaller skulls compared to the large *Artibeus*, they may compensate for their small skull size by enhancing their mechanical advantage to consume hard figs that the large *Artibeus* species are able to eat as a result of their larger skull size. Alternatively, morphological variation in *Artibeus* spp. may be related to allometry or drift rather than function. Future work evaluating skull shape variation in *Artibeus* spp. across its geographical range in connection with field studies demonstrating which fruits are eaten at each locality will help to establish a form-function relationship. Such work will also uncover the role of behavioral and dietary plasticity in *Artibeus* spp. to allow a better understanding of how these factors and the morphological diversity demonstrated in the present study may have shaped the evolution of the *Artibeus* species complex.

## Sexual dimorphism

In addition to interspecific morphological variation, this study showed that both sexual size dimorphism and sexual shape dimorphism were present in lateral and ventral views in the *Artibeus* species complex as a whole. *A. lituratus* had significant sexual shape and size dimorphism in lateral view, but not in ventral view. In lateral view, the females had a lower skull profile than males (Fig. 7A). Further, *A. lituratus* females were larger than males (Figs. 7C, 7D). Female *A. obscurus* have also been noted to be larger than males (*Brosset & Charles-Dominique, 1990*; *Eisenberg & Redford, 1989*; *Simmons & Voss, 1998*). As sexual size dimorphism is present in lateral view for *A. lituratus* and is only marginally significant in ventral view (Table S6), even though the two size metrics strongly agree, the two-dimensional view used to calculate centroid size may have a bearing on significance. Further, since sexual dimorphism was assessed in the interspecific data at the genus level as a result of sample size limitations, it is not possible to determine which specific *Artibeus* species were sexually dimorphic (with the exception of *A. lituratus*). Given the low $R^2$ value associated with sex in the interspecific data, it is likely that some species display sexual dimorphism while others do not. A deeper analysis of other *Artibeus* species with large sample sizes, similar to the one performed here for *A. lituratus* will help clarify sexual dimorphic trends for other *Artibeus* species.

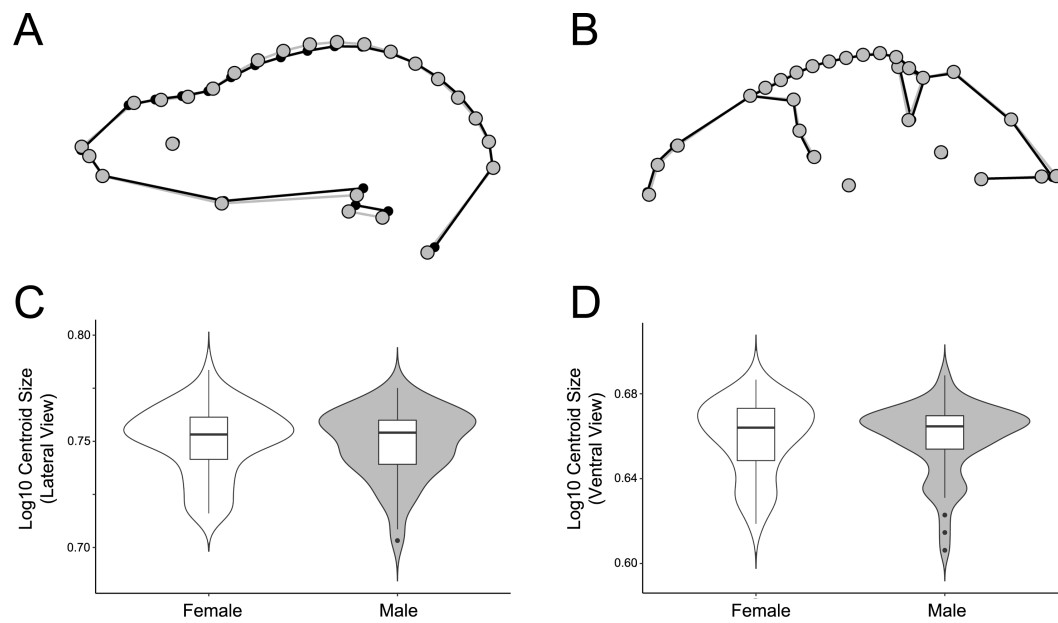

**Figure 7 Comparison of male and female shape and size in lateral (A, C) and ventral (B, D) views.** For shape wireframes, males are gray and females are black. Wireframes are magnified by three times to accentuate differences, though these differences are quite minor in general.

The significant sexual dimorphism found here has implications for behavioral ecology and niche partitioning in the *Artibeus* species complex. Behavioral differences between sexes have been noted in *A. jamaicensis* (*Kunz & Diaz, 1995*). Males forage primarily near their roosts, allowing them greater time for roost defense, while females forage greater distances and for longer periods of time (*Morrison, 1979*; *Morrison & Morrison, 1981*). These differing strategies may affect cranial morphology. For example, females have the opportunity to seek out higher quality food since they do not have to rely on the food near their roosts. Thus different feeding behaviors in each sex might have an impact on sexual size dimorphism in cases where males and females are feeding on different quality fruits. For example, body mass has been correlated with fruit size in *A. jamaicensis* and *A. lituratus* (*Ortega & Castro-Arellano, 2001*). Since fruit hardness and skull shape have also been correlated in a wide variety of phyllostomids (*Freeman, 2000*; *Nogueira, Peracchi & Monteiro, 2009*; *Dumont et al., 2012*; *Hedrick & Dumont, 2018*), these differences may also lead to the sexual shape dimorphism found in the present study's data if males and females choose fruits of different hardnesses. In lateral view, male *A. lituratus* have a slightly taller, more dome-shaped skull with a more pronounced sagittal crest and shorter rostrum (Fig. 7A), which suggests that they are able to consume somewhat harder fruit given a somewhat higher mechanical advantage for the temporalis muscle. The shape of the zygomatic arch in ventral view is identical in male and female *A. lituratus* indicating no difference in their masseter muscle (Fig. 7B). The presence of a stronger sagittal crest on male skulls leads to the potential hypothesis that male *A. lituratus* are capable of higher bite forces and eating harder fruits than females (*Santana, Dumont & Davis, 2010*), although this needs

to be thoroughly evaluated using bite force data. Other alternative hypotheses related to higher sagittal crests in males may include biting during male-male combat or tent-making. Alternative to the functional-related hypothesis that is proposed, sexual selection for larger size in females may play a role in skull shape. Female *A. lituratus* have larger skulls than males (Figs. 7C, 7D), which is common among bat species where the female is typically larger so she can maintain homeothermy during pregnancy and fly while carrying her pups while they are nursing (*Williams & Findley, 1979*). Larger size in females may lead to a less developed sagittal crest in comparison with males as they may be able to eat relatively harder foods by virtue of larger body size. More work on potential dietary differences between male and female *Artibeus* species will need to be done in order to better elucidate these results.

### Intraspecific and geographical variation in *Artibeus lituratus*

*Artibeus lituratus* has an expansive range from Mexico to Argentina. This study demonstrates substantial geographical variation in *A. lituratus* skull size and shape with specimens forming country locality clusters in morphospace (Figs. 6A, 6C). However, the data do not have an obvious pattern of clinal variation, which is demonstrated most clearly by the UPGMA results. In lateral view, two main groups formed using UPGMA of shape data, one containing Peru and Belize, and the other with Paraguay, Argentina, Trinidad, Costa Rica, and the two Mexican localities. The ventral view data had a similar core with the two Mexican localities, Costa Rica, and Trinidad grouping together, but the other country groupings changed (Figs. 6B, 6D). Belize being outside of the core group of northern countries in spite of being one of the northernmost localities sampled highlights the lack of clinal structure in the shape data. The pairwise results also demonstrate some lack of consensus between the lateral and ventral views for many countries. For example, Belize is significantly different from all countries other than Costa Rica, Trinidad, and Paraguay in lateral view, but is significantly different from all countries other than Costa Rica and the two Mexican localities in ventral view. In this example, the ventral view suggests a geographical cluster of northern localities, while the lateral view suggests no geographical clustering. Size variation in *A. lituratus* appears to generate two groups, north and south. Specimens from Belize, Mexico, and Costa Rica had the smallest average centroid sizes among specimens sampled while specimens from Argentina, Trinidad, Peru, and Paraguay had the largest (Table 4). General clinal size variation has been reported in other bats as well (*Nagorsen & Tamsitt, 1981*; *Owen, Schmidly & Davis, 1984*; *Storz et al., 2001*) and a wide variety of other species (birds, *James, 1970*; mammals, (*Koch, 1986*); insects, *Chown & Gaston, 2010*). In spite of this overall grouping, there was no clear clinal pattern within either the northern or southern groups.

Only in the extreme northern and southern localities did a consistent shape pattern emerge. In lateral view, Argentina is significantly different from all other localities, and in ventral view it is significantly different from all northern localities (Mexico, Belize, Costa Rica), while not significantly different from South American localities (Paraguay, Peru, Trinidad). Similarly, the two Mexican localities are significantly different from all localities other than Costa Rica in lateral view while they are significantly different from all South

American localities in ventral view and not different from the other northern localities. This strongly suggests that the *A. lituratus* range only has substantial differences at northern and southern extremes and otherwise does not exhibit a clinally graded morphological spectrum for cranial shape. This demonstrates that analyzing specimens from across a species' range will grant a better understanding of the shape disparity within that taxon, especially in taxa with large ranges such as *A. lituratus* where significantly different morphologies at the extremes of the species' range may be missed.

The lack of an obvious geographical pattern in these data may be due to differences in environmental conditions at the localities where the bats were collected rather than spatial differences. The present study did not assess environmental variables, but instead looked for general skull shape and size differences in relation to geographical distance. In *Eptesicus fuscus*, *Burnett (1983)* found that both climatic factors and geography affected morphology, with geography tending to be a better predictor for wing shape and environment tending to be a better predictor for skull shape. Using linear morphometrics, *Marchan-Rivadeneira et al. (2012)* found that skull size variation in *A. lituratus* was tied closely to environmental variables. Factors such as seasonality and precipitation were tightly correlated with skull size. *Stevens, Johnson & McCulloch (2016)* examined wing shape variation in *A. lituratus* in the Atlantic forests of Paraguay and Argentina and found that environmental variables accounted for 75% of variation across sites. They suggest that this is due to local adaptations in wing morphology, which enhance maneuverability in some environments. Therefore, there may be a larger pattern among bats that both wing shape and cranial shape are more closely tied to environmental factors than geographical distances. Further, taxa in the Amazon are often considered to present more of a mosaic of characters rather than a clear pattern, such as clinal variation (*Bates, Hackett & Cracraft, 1998*; *Ferreira et al., 2014*). This may be the case for *A. lituratus* crania.

Genetic differences across the wide *A. lituratus* range may also have led to the north-south size differences that are captured in this study. Two subspecies are sometimes recognized (*A. lituratus intermedius* and *A. lituratus palmarus*), though are also often considered different at the specific level (*A. lituratus* and *A. intermedius*) (*Simmons, 2005*; *Redondo et al., 2008*). Unfortunately, the subspecies identification was not available for the specimens included in this analysis so this could not be assessed. Regardless of subspecies identification, genetic drift may be a factor generating the difference in size between northern and southern specimens of *A. lituratus* as well. Given this study's clear documentation of the range of individual variation present in *A. lituratus*, a more specific study examining skull shape and size in habitats with different environmental variables in the northern and southern parts of the *A. lituratus* range would allow for more concrete conclusions.

## CONCLUSIONS

The *Artibeus* species complex is a widespread, highly diverse lineage, with many species that live in sympatry. Being able to quantify variation and differentiate species morphologically is a step towards understanding the diversification of the *Artibeus* species complex and the factors that drove its radiation more generally. Morphological variation was not only

evident both between the small *Artibeus* and large *Artibeus* groups, but fine-scale size and shape variation was also evident within both groups. Shape data were significantly correlated with phylogeny, likely due to the strong clustering of species within the small and large *Artibeus* groups. Further, given that *Artibeus* feed on different fruits throughout the year, it is possible that a combination of morphological and behavioral plasticity has led to the modern diversity of the group. This study demonstrated morphological separation between males and females, perhaps due to these behavioral drivers; however, sex accounted for a small percent of total shape variation. Finally, geographical variation is a clear factor governing intraspecific variation in *A. lituratus*. This variation is not obviously driven by clinal factors and likely relates to site-level environmental variables or drift rather than broad geographical distances. Being able to accurately delineate and discriminate among species is critical for both taxonomy and for conservation initiatives (*Fernandes et al., 2009*) and quantifying inter- and intraspecific variation across a broad geographic range grants insight into how morphological variation in a group is structured, allowing for second-order analyses. Future work on the ecological and behavioral variables that have led to the documented morphological variation among species in *Artibeus* will help to elucidate the evolution of this successful group of bats.

## ACKNOWLEDGEMENTS

I thank Jacob Esselstyn (LSU) for access to specimens and help in the Louisiana State University Museum of Natural History collections. I thank Sam Cordero, Andrea D. Rummel (Brown Univ.), V. David Munteanu (Clemson Univ.), and Samantha L. Cox (Univ. of Penn) for comments and insights on earlier versions of this manuscript. I thank Ilse Garcia Romero and Liliana Dávalos (Stony Brook University) for writing and editing the Spanish language abstract. Finally, I thank Daniel Silva (editor), Kara Powder, Fabio Machado, and three anonymous reviewers for reviews that have greatly improved this manuscript.

### Funding

The author received no funding for this work.

### Competing Interests

Brandon P. Hedrick is an Academic Editor for PeerJ.

### Author Contributions

- Brandon P. Hedrick conceived and designed the experiments, performed the experiments, analyzed the data, prepared figures and/or tables, authored or reviewed drafts of the paper, and approved the final draft.

### Data Availability

All data are available in the Supplemental Files.

## Supplemental Information

Supplemental information for this article can be found online at http://dx.doi.org/10.7717/peerj.11777#supplemental-information.

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
