# Peer review of "Inter- and intraspecific variation in the Artibeus species complex demonstrates size and shape partitioning among species"

_PeerJ, doi:10.7717/peerj.11777_

## Round 0.1 · original submission · Major Revisions

Dear Dr. Hedrick,

After the review of four different reviewers, I consider that your manuscript may be accepted for publication in PeerJ after your perform changes to your text. Of the four reviewers, three of them indicated major reviews, whereas reviewer #2 indicated minor reviews.

When you resubmit your research, please also provide a resubmission letter, explaining to the reviewers all the changes that were done or explaining why you did not perform some of them as well.

Sincerely,
Daniel Silva

Reviewer 1 ·

Basic reporting

The paper is well written. However, it is primarily focused on the system. Therefore it might be a better fit for a taxon-specific journal. I recommend providing more background information in the Introduction relevant to the question(s) instead of focusing exclusively on the system. By making these changes, the paper will be more attractive for a wider audience and possibly more suitable for PeerJ readers. Also, I wonder if the author could make hypotheses that go beyond testing previous findings. Thus, the author would focus on the novelty of the results presenting in the paper, their new hypotheses and interpretations.

Experimental design

The study aims to assess the extent of the inter- and intra-specific variation in the skull’s size and shape of Artibeus. Although the author did analyze the data to test for the primary goal of the study, some information is required to validate the accuracy of data collected. In geometric morphometrics, it is well known, the need to account for measurement error, which is regularly accounted for by digitizing and photographing images at least twice. It is not mentioned that this was accounted for, so it is important to mention if it was (or not) the case. Also, in morphometrics, evaluating the degree of differences between sexes is traditionally recommended to be assessed before conducting any further analysis, particularly in a group for which previous studies have reported sexual dimorphism. The author did evaluate for differences between sexes. Still, it is unclear why the author did not partition the sample or run analysis separately by sex after finding statistical significance between males and females in the Procrustes ANOVA. One possibility for not doing this is the limited sample size in each of the two datasets analyzed. If so, it has to be mentioned.

Validity of the findings

The author overestimates the scope of the data used to conduct the analyses by referring to it as a “large sample.” Given the wide distribution of most of the species included in the analyses, it is unclear how the data could reflect the scope of the inter- and intra-specific variation in the group. More importantly, analyzing the level of intraspecific variation within a single species in the genus, at what scale, can be meaningful to understand overall patterns of variation in the group. Finally, the Discussion section might need to provide a more extended exploration of how the findings can be used to make inferences about patterns of variation in the group beyond the actual distribution of the species in the morphospace. Therefore, there is a need to link the findings to mechanistic explanations associated with environmental, behavioral, or phylogenetic clustering that could be used to interpret the Results.

Additional comments

- It is recommended to expand the justification to conduct the study in the Introduction. Otherwise, the whole motivation is primarily focused on the system and not necessarily on the processes that can be driven differences in shape among species and within them. For instance, why do we expect differences in shape and size to arise? How has morphology been reported to show “higher” variation between species that occupy large geographic ranges? How can ecology play a role in this?
- Line 129: In Figure 1, the author shows a phylogeny based on data presented by Rojast et al. (2016), which, although it does not reflect the most recent revision in the group (see Hoofer et al. 2008; Larsen et al. 2015; Redondo et al. 2008; York et al. 2019), it should be properly cited and explained on the legend to understand better how the phylogeny was generated and add the methods used to plot it in the corresponding section.
- Line 35. The author refers to the localities included in the intraspecific analysis, plotted in Figure 2. However, it is unclear by just seen the map the location of the sites where the specimens examined where collected. It is recommended to mark each of the unique locality points included in the study, instead of highlighting countries that are just geographic constructs that in nature might not reflect the actual distribution of species (or subspecies in the case of the intraspecific dataset). Also, it would be good to see the location of the sites for the interspecific analyses too, and ultimately providing a graphical representation of the geographic distribution of each of the species analyzed will allow assessing better how the sampling can capture the scope of the variation across the geographic range of the species, instead of potentially increasing the geographic differences among sites just by an effect of geography.
- Line 167. The author mentions that due to “sample size limitations,” sexual dimorphism in Artibeus species was assessed as a whole, instead of within species as it was done in A. lituratus. How can this influence the outcomes?
- Line 175. The author mentions that a phylogenetic Procrustes ANOVA was not conducted because of the need to summarize all the variation within each group using mean values and this would “loss” differences accounted by sex. Traditionally, when sexual dimorphism is evaluated, and there is statistical support to validate it, further analyses to evaluate the scope of variation in the sample must be conducted separating males and females. Did you consider doing that? It might be important to do this, especially if one of your goals focuses on testing this potential source of variation, which you found evidence for in your Procrustes ANOVA (line 253), but it seems that the author disregarded it by combining the data for the rest of the analyses.
- Line 260. The author found that differences among species in the morphospace are accounted for by phylogeny. However, this is not discussed later in the manuscript. How can this be interpreted? What kind of inferences can be made based on these results?
- Figure 6. How do you explain the differences in the arrangement of localities in the UPGMA cluster?
- Line 296. It is stated in the paper that the “overarching goal of this study was to understand better the relationship between intraspecific and interspecific variation and morphological differentiation based on a large sample.” How can intraspecific patterns of variation in A. lituratus reflect the scope of variation in the group within species? Given the wide range of distribution of most of the species included in the analysis, how the limited sample size to evaluate inter- and intra-specific variation can influence the outcomes?
- In the Discussion section, the author should avoid being too descriptive with the findings and instead focuses on their interpretation and possible inferences.
- How the differences accounted for by sex can be explained? How could geography play a role in the scope of variation reported between species and within them?
How can changes in the shape and size of the skull impact species' fitness in this group?
- What is the interpretation of the level of variation observed between species?
- Line 387. Is the intraspecific arrangement of populations in A. lituratus associated with the currently recognized subspecies in the group?

Reviewer 2 ·

Basic reporting

Basic reporting is all fine. This is an interesting and well written manuscript.

Experimental design

The only concern that I have with the experimental design are the identities of the specimens. I do not believe that the bats in the LSU mammal collection have been curated since the 1970's. The Taxonomy of Artibeus has been very fluid since then. Did the author accept the species names on the tags of the specimens or did they independently verify the ID of each individual? If they did not independently ID specimens, when was the last time that the Artibeus collection was curated? Based on the data provided by the author, there is no way to establish even where the specimens come from. I would like to see a specimens examined section added to the manuscript and in that speciments examined, the location of each specimen be disclosed.

Validity of the findings

All underlying data have not been provided. The authors need to provide the raw landmark data. It appears that they make the derived data (PC's) available, but not the raw landmarks. As stated in the Intro, Artibeus is not the most species rich genus of Phyllostomidae...Sturnira is.

Additional comments

Nice manuscript, just concerned about the specimen ID's.

·

Basic reporting

Clarity of hypotheses and gap in knowledge: This work is described (L95-100) as a re-assessment of interspecific variation in this same group, with a different analysis method. Could you expand on why you might expect different results with this new method or limitations of previous methods? This would also help define a clear hypothesis, which is currently lacking, though predictions are laid out in lines 106-225. The manuscript currently doesn't define a clear gap in knowledge.

Presentation and visualization of data (see also comments under #3): (1) I appreciate that there were 2 different sets of color for inter-and intraspecific analyses. However, I found myself trying to find colors from Fig 1 in the map in Fig 2 and wondering, e.g., if the light green colors were supposed to match. A few suggestions to try to distinguish these two analyses more: Perhaps different symbols could be used in figures 4-6? How about moving Figure 2 to directly before Figure 6? (2) Rather than a boxplot, using a violin plot, adding individual points to the boxplot, or using another alternative that shows the distribution of data points would avoid known shortcomings of a basic box plot and more fully detail your data in Figure 5. (3) Have color sets used been tested regarding readers with red-green colorblindness or those printing in black and white?

Raw data: Stated as provided to the journal. Perhaps it would be more accessible if posted in dryad or other repository that have clear policies about archiving as well?

Experimental design

Effect of size: (1) I appreciate that the allometric trend in this data is assessed, particularly given the size differences between specimens and statistical association of size on shape. Is there a reason size correction was not done to generate allometry free shape principal component measures, a function that is built into geomorph, rather than assessing via ANOVA and common allometric component? Was it possible to assess overall length of animals, or were only crania available? This may be particularly important as PC1 explains both size and shape. Shape differences (which seemed like the key question in the analysis) may be more distinct once size is removed from the variation. (2) What are size differences in intraspecific analysis in Figure 6? Based on ANOVA analyses in L280 and L283), it sounds like this is a contributing factor to variation amongst countries, but this is not addressed within the presented data in Figure 6.

Choice of species and phylogenetic corrections: (1) While I appreciate wanting to preserve the sexual dimorphism component (L174-177), is there a reason phylogenetic correction Procrustes ANOVAs were not performed to verify species-level signals? That is, one analysis to verify this was not causing a bias in conclusions about species, and another analysis to more specifically look at the influence of sex on shape? (2) Are the 11 species chosen representative of Artibeus cranial variation?

Validity of the findings

Illustration and inclusion of data to assist readers in reaching conclusions: (1) I found myself wishing for more visualization of the data in Figures 4-5. For instance, a warp of mean shape for each species, a violin or other plotting of range of shape by species or species and sex, or even outlines on the morphospace to visualize distinctions between species. These types of figures would help readers make their own assessments of distinctions between shape in small and large Artibeus groups (results L201-203), and other descriptions of data in the results section. As is, this is difficult for the reader to parse easily, without matching the full species name with abbreviations in the figures, then searching for the colors in the morphospace. (2) What are the shape differences on PC2? This warp is not illustrated. This is especially interesting as species in the small Artibeus group were separated in PC2 scores. Additionally, warps of shape variation are not provided at all for intraspecific analyses of A. lituratus. (3) Sex was stated as an important variable (L174-177, results in L252-259, based on statistical analyses, and a L359-385 in discussion), but this is not illustrated in figures. Perhaps this could be done with symbol size or filled/unfilled in Figures 4-6? Alternatively (and perhaps preferably for clarity), should there be a figure comparing sexes if this is an important conclusion?


Relation of shape changes to niche and foraging: (1) A primary discussion point was how these shape changes related to feeding niche (L24-25 and L26-31 of abstract, L312 “strongly suggests niche partitioning of species,” and diets are discussed in L346-357). I believe there is extensive information about what these species eat. Could this be incorporated and tested? Alternatively, this is presented as a supported conclusion in the abstract, rather than speculation, as it appears in the discussion. This discrepancy needs to be addressed. (2) Literature is cited in L380-381 about the relationship of skull shape to fruit hardness. Given these and other literature related to function of different skull shapes, can more details about feeding be addressed in the discussion? That is, rather than concluding that there are “different niches” could there be a discussion of the impact of shape on feeding mechanics? This would require further detailing of shapes in the results, as the shapes detailed are currently pretty scant. E.g. “shorter rostrum” in L210, or no visualization of the shapes between males and females or warps for intraspecific analyses.

Alternative explanations not fully discussed: Foraging is presented as the explanation for all shape variation. Are there additional speculations about the functional impacts or driving forces that cranial variation? For instance: (1) What is the relationship between geography and phylogeny across species, and more specifically within A. lituratus? That is, is there an alternative explanation (genetic variation) that could explain the differences in morphology, rather than diet? (2) For intraspecific analyses, it is mentioned that there are no patterns of geographical variation (L307-308). However, it does not appear that these designations are included in figures or Table S5 to support that conclusion. That is, can a non-specialist who doesn’t know the various collection locations also reach this conclusion? (3) Could sexual selection explain sexual dimorphism rather than sexes choosing fruits of different hardness?

Other minor comments:
--What is shape on the Y axis in Figure 5? PC1 score?
--L232 Should “dorsal PCA” be “lateral PCA”
--Are descriptions of shape differences being minor in interspecific analyses (L 273-274 and L278-279) based on qualitative assessments or can this be quantitatively or statistically assessed?
--L305 Should this read “…contrary to previous studies…”

Additional comments

This study assesses both inter- and intraspecific variation in cranial morphology within the adaptive radiation of bats. While clearly written and methodically sound, additional analyses, specifically size and phylogenetic corrections, as well as testing of foraging speculations, would greatly strengthen conclusions or allow current speculations within the abstract to be appropriately assessed. Further illustration and visualization of data as I have suggested above should be improved upon before acceptance and would not require additional review. However, additional analyses such as size correction or diet analysis may alter conclusions drawn or allow a new line of more detailed conclusions to be drawn, which may necessitate another review.

·

Basic reporting

The manuscript is well written and follows the PeerJ standards in terms of structure. Figures are generally well labeled and illustrate the results correctly. Some graphical elements are lacking, specifically the representation of shape deformations grids on figures 4 (PC2), 5 (CAC), and 6 (PC1 and 2). Tables 1 and 2 legends lack some relevant elements to properly understand them, namely the statistical terms (Df, SS, MS, etc.). Data provided in the supplementary material is enough to replicate figures and is easily accessible and adequately explained. In terms of general structure, the manuscript seems adequate.

Experimental design

The introduction adequately highlights the paper's conceptual background and identifies the gap in the knowledge that the paper expects to fill. Specifically, the paper seeks to perform an exploratory analysis of the inter and intraspecific shape variation in the Artibeus species complex, a genus known for its conserved morphology. Sample sizes are more than adequate for the proposed investigation. The methodology is designed to investigate if species differ and if different populations of a more widely distributed species also differ. The methods used are straightforward and rely on linear models to examine these issues and the potential presence of sexual dimorphism and allometry. Methods are satisfactorily explained in most cases, and results are transparently presented and discussed.

Findings from this manuscript stem directly from the investigation of the available dataset. The result that Artibeus species are more different than previously imagined is new and exciting and points to the possibility of local adaptation and niche partition. The evaluation of the intraspecific dataset shows that local adaptation is also a possibility in A. lituratus. Sexual dimorphism also suggests that ecological differences between males and females of the group might express in terms of sexual dimorphism.
Nevertheless, the paper neglect some aspects of this morphological diversity. For example, besides broad stroke differences described on the PC1 for both views and some aspects of sexual dimorphism, little is known how species diverge. Descriptions of PC2 are not given, nor shape deformations associated with CAC. This absence is even more problematic as PC1 and the allometric vector seems to share some common variation. Results show that species differ after accounting for differences in size, but we cannot say precisely how. This problem can be solved by amending the current methodology (correctly describing and showing differences between species and sexes) or using new methods (see comments to the author).

Validity of the findings

Results from the analysis are adequately explained, and the analyses corroborate the conclusions. The results are new and can be interesting for anyone interested in either taxonomy, ecology, or evolution of the group. The only issue with the conclusions is the supposed lack of clinal variation in the intraspecific dataset. While I agree that there is no perfect geographical pattern, it is not possible to discard the existence of clinal variation without explicitly investigating it (see comments to the author). While broad patterns of shape variation might not be geographically determined, smaller scale-variation might be geographically structured. A proper examination of geographical variation might help to solve this issue. Furthermore, I feel that other aspects of the methodology and discussion can be improved to support the conclusions better and improve readability (see comments to the author).

Additional comments

I found the paper reasonably well designed, and the methodology is good enough to justify the discussion. However, the paper touches on a couple of topics that can be better explored with available methods that put the discussion on a sounder footing. In that regard, anything that I suggest below in terms of methodology should be viewed as ways to improve the paper and do not reflect any flaws of the paper. I will list topics and what methods can be used to improve the results and conclusions.

Data processing- The author employs the standard method of evaluating each view independently, which entails that all methods have to be replicated twice. However, some recent papers have suggested ways to assess different views/configurations on the same dataset (Antonio et al. 2019. Seeing the wood through the trees. Combining shape information from different landmark configurations. Hystrix-the Italian Journal of Mammalogy, 30(2), 157–165. http://doi.org/10.4404/hystrix–00206-2019; Collyer et al. 2020. Making Heads or Tails of Combined Landmark Configurations in Geometric Morphometric Data. Evolutionary Biology 47, 193–205. http://doi.org/10.1007/s11692-020-09503-z). The author should contemplate using this methodology for two reasons. One, it might improve readability by simplifying methods, results, and discussion. Two, by analyzing the skull shape as a whole, different patterns might arise. I suggest using the methodology exposed in Antonio et al. (2019) as a starting point (but keeping in mind comments made by Collyer et al. 2020). Furthermore, regarding semi-landmark sampling on the zygomatic on the ventral view: the number of points seems excessive. The same variation can probably be sampled with fewer points (even half of the points used).

Niche partitioning- Most of the justification for the argument that niche partitioning might be guiding differences in species relies on the fact that species were significantly different on the non-parametric MANOVA tests. However, MANOVAs tests for the differences in average, not overlap between groups, which would be more descriptive of ecomorphological niche partition. To directly evaluate morphospace overlap, the author can employ two different but similar methodologies. One is to use Linear Discriminant Analysis to calculate between species reclassification rates as a measure of overlap (e.g., Machado, F. A., & Teta, P. 2020. Morphometric analysis of skull shape reveals unprecedented diversity of African Canidae. Journal of Mammalogy, 26, 32–12. http://doi.org/10.1093/jmammal/gyz214). Since LDAs require matrix inversion, a dimensionality reduction is always advised before using this method (see Miranda et al. 2017. Taxonomic review of the genus Cyclopes Gray, 1821 (Xenarthra: Pilosa), with the revalidation and description of new species. Zoological Journal of the Linnean Society, 183(3), 687–721.http://doi.org/10.1093/zoolinnean/zlx079). One alternative is to use the package nicheROVER (Swanson et al. 2015. A new probabilistic method for quantifying n‐dimensional ecological niches and niche overlap. Ecology 96.2: 318-324.), which do not require matrix inversion nor equality between covariance matrices between groups, but still uses multivariate normal distributions to model populations. If that assumption is not reasonable, one alternative is "hypervolume" (Blonder et al. 2014. "The n‐dimensional hypervolume." Global Ecology and Biogeography 23.5: 595-609.), which allows for the calculation of more complex hypervolumes and overlaps.

Phylogenetic signal and evolutionary patterns- A high phylogenetic signal by itself can be consistent with both neutral evolution (Brownian motion) and early burst scenarios. The author might consider employing a Diversity Through Time to supplement their phylogenetic signal analysis (Murrell, D. J. (2018). A global envelope test to detect non-random bursts of trait evolution. Methods in Ecology and Evolution, 9(7), 1739–1748. http://doi.org/10.1111/2041-210X.13006). DTT might give some insight into the process leading to the diversification of a given phenotype (e.g., Segura et al. 2020. Integration or Modularity in the Mandible of Canids (Carnivora: Canidae): a Geometric Morphometric Approach, 1–13. http://doi.org/10.1007/s10914-020-09502-z). Stable patterns of disparity are consistent with stability, while quick drops of disparity over time are associated with bursts of diversification. This might help to better discuss evolutionary scenarios that might have shaped the diversification of the group.

Geographical variation- One possibility to evaluate geographical patterning in the intraspecific dataset is through the use of linear models with geographical variables. This can be done in two main ways: trend surface analysis, which is a regression of geographical coordinate derived variables on shape, and eigenvector mapping, that decomposes spatial structure in new eigenvectors that represent geographical variation in multiple scales. See Cardini et al. (2010. Biogeographic Analysis Using Geometric Morphometrics: Clines in Skull Size and Shape in a Widespread African Arboreal Monkey. In A. M. T. Elewa (Ed.), Morphometrics for Nonmorphometricians (5 ed., Vol. 124, pp. 191–217). Berlin, Heidelberg: Springer Berlin Heidelberg. http://doi.org/10.1007/978-3-540-95853-6_8) and references therein. Another more straightforward and less powerful alternative is employing a Mantel test on Procrustes distance and geographical distances. The absence of a signal on any of these analyses will support the conclusion that there is little or no geographical structure on the data.

Still, regarding intraspecific variation, one question that is not explored is how similar it is to the interspecific variation in magnitude and pattern. Magnitude could be investigated by calculating disparity for both intraspecific and interspecific datasets using the function morphol.disparity() from the geomorph package. To examine if intra and interspecific variation patterns coincide, the author can evaluate the concordance between the leading PC of intraspecific data and the PC of a phylogenetic covariance matrix (see Segura et al. 2020; Revell, 2009. Size-Correction and Principal Components for Interspecific Comparative Studies. Evolution, 63(12), 3258–3268. http://doi.org/10.1111/j.1558-5646.2009.00804.x; Slater and Friska .2019; Hierarchy in adaptive radiation: A case study using the Carnivora (Mammalia). Evolution, 73(3), 524–539. http://doi.org/10.1111/evo.13689; Parsons et al. 2020. Skull morphology diverges between urban and rural populations of red foxes mirroring patterns of domestication and macroevolution. Proceedings of the Royal Society of London. Series B, Biological Sciences, 287(1928), 20200763–10. http://doi.org/10.1098/rspb.2020.0763). A simple vector correlation or angle comparison could be enough to direct discussion in this regard. This could help to evaluate if geographical variation within-species follows the same pattern that between species evolution, suggesting common ecological drivers of both levels of morphological adaptation (see Schiaffini et al. 2019. Geographic variation in skull shape and size of the Pampas fox Lycalopex gymnocercus (Carnivora: Canidae) in Argentina. Mammalian Biology - Zeitschrift Fur Saugetierkunde, 97, 50–58. http://doi.org/10.1016/j.mambio.2019.04.001). Vector correlation might also be interesting to evaluate if patterns of divergence (PC1) and allometric patterns coincide (e.g., Machado and Teta, 2020). Vector correlation and angle comparisons can be implemented with the function angleTest() from the package Morpho.


Minor comments

L74- This is a strong paragraph to introduce your model system. Consider starting the paper with this one instead.
L95- Rossoni et al. (2019) is not a geometric morphometric investigation.
L174-177- A phylogenetic ANOVA wouldn't be appropriate for an investigation into between-species differences. This phrase is not necessary.
L186-189- Why not simply call "countries" "populations" and name them according to countries? Also, the investigation of "within country variation" makes little sense, as countries are arbitrary geographical categories. You don't need to justify why you have two Mexican samples.
L257-258- It is not clear from the text if this passage refers to the analysis of shape or size. If the author refers to shape here, then the interaction between those factors does not mean what is stated. Instead, it means that allometric relations change with species and with each sex within each species. If this passage refers to size, it needs to be rephrased to better express that.
L308- It is unclear if this sample is adequate to identify differences between Atlantic Forest and the Amazon. Samples seem to be marginal to these biomes, so this statement appears unnecessary.

---

## Round 0.2 · Minor Revisions

Dear Dr. Hedrick,

After two independent reviews of the new version of your MS, both reviewers agreed the text was very much improved, with one of them being a new reviewer. The new reviewer suggested minor reviews, and as soon as they are addressed, the manuscript will be accepted.

Congratulations,
Daniel Silva.

·

Basic reporting

No comment

Experimental design

No comment

Validity of the findings

No comment

Additional comments

Author was very responsive to my and other reviewer comments. All previous concerns and comments have been addressed in this revised manuscript.

Reviewer 5 ·

Basic reporting

Greetings,

I have reviewed Dr. Hedrick’s manuscript, which presents a study of the variation in skull shape within and among Artibeus species. Additionally, I have looked at the review files that were part of the submission. The reviewers’ comments from the first round of reviews were extensive and thorough and I largely agree with the points they made, and the author seems to have addressed them appropriately in this version. As a result, I only have suggestions that involve text edits to further improve the manuscript.

Abstract:
L19-20: This sentence needs work. The fact that species coexist means that they have not competitively excluded one another and therefore there is niche partitioning. This can be achieved through differences in non-morphological traits (e.g., size, behavior, acoustics). What the dearth of variation may mean is that skull shape changes may have not evolved as a result of competition.
L25: replace “niches” with “ecological resources”

Introduction:
L68-69: All species live in sympatry with related species.
L25: also vertebrates.
L95-105: this paragraph could be greatly condensed. Geometrics morphometrics is now over 30 years old, so presenting it as a revolutionary tool seems outdated and unnecessary. I recommend deleting this text starting with "Caliper-based linear morphometrics..."
L111-113: this does require further explanation; why would geometric morphometrics actually allow to capture these differences? It is because it allows to disentangle shape from size variation?
L117: delete “rather than linear morphometrics”
L117: how many species are there in the Artibeus spp. complex? Clarify what portion of the variation is examined here.
L119-120: “which may suggest differences in niche partitioning” should be balanced with a non-adaptive explanation (e.g., “or non-adaptive processes such as drift”). In general, the manuscript strikes me as very adaptationist – not all morphological differences are due to adaptation, as some of the results illustrate.
L122: what is the prediction and justification for examining sexual dimorphism? This should be added.
L128: add justification why this species was selected for the intraspecific study.
L132: underlying causes are discussed, not assessed.

Methods:
L163-165: clarify here and in figure legend -- were the semilandmarks placed evenly spaced along the curve? Add them to the curves in Fig. 2.
L194: means of what?
L196: clarify this is Kmult.
L220: add text to clarify why country was used as a proxy for locality, since this is unusual.
L224: the use of “population” here seems problematic. Replace with “individuals in portions of the geographic range (countries)…”

Results:
L315-318: this could favor non-adaptive hypotheses – a strong phylogenetic signal in size differences and thereby shape via allometry.
L341-342: it’s unclear how these tests were conducted and what is being shown in the table (what are the coefficients shown?). Clarify in methods and Table legend.

Fig 4: change large Artibeus symbols to triangles for readability and consistency.

Discussion:
The discussion section largely recapitulates the results, and is still narrowly focused on phyllostomid bats. For a journal like PeerJ, the results should be discussed in the context of the broader literature on morphological variation, sexual dimorphisms, etc.
L360: replace “successful” with “diverse”.
L381: one possible explanation, if these differences result in performance differences, resource use, etc.
L392: replace “workers” with “studies”.
L416-417: this sentence is very vague; state what you mean more specifically.
L418-419: this sentence suggests that plasticity selects on morphology, but the studies cited do not demonstrate this.
L420-422: this reference is not in the reference list. Did this study actually couple quantitative dietary data with behavioral data and modeling of morphological evolution (or paleontological studies) to demonstrate this?
L433-436: this is assuming the variation leads to functional consequences of importance to access food resources - size may simply override these though.
L444: or other factors that affect body size may “drag” the evolution of skull size and shape with it.
L455-456: unclear what is meant here by “the way size is calculated”
L478-479: how about defense, fighting other males, tent making?
L512-513: there is a rich literature on this for mammals and other vertebrates, expand the scope beyond this narrow group.
L568-569: what is the purpose of this sentence?

Experimental design

No comment.

Validity of the findings

No comment.

Additional comments

Greetings,

I have reviewed Dr. Hedrick’s manuscript, which presents a study of the variation in skull shape within and among Artibeus species. Additionally, I have looked at the review files that were part of the submission. The reviewers’ comments from the first round of reviews were extensive and thorough and I largely agree with the points they made, and the author seems to have addressed them appropriately in this version. As a result, I only have suggestions that involve text edits to further improve the manuscript.

Abstract:
L19-20: This sentence needs work. The fact that species coexist means that they have not competitively excluded one another and therefore there is niche partitioning. This can be achieved through differences in non-morphological traits (e.g., size, behavior, acoustics). What the dearth of variation may mean is that skull shape changes may have not evolved as a result of competition.
L25: replace “niches” with “ecological resources”

Introduction:
L68-69: All species live in sympatry with related species.
L25: also vertebrates.
L95-105: this paragraph could be greatly condensed. Geometrics morphometrics is now over 30 years old, so presenting it as a revolutionary tool seems outdated and unnecessary. I recommend deleting this text starting with "Caliper-based linear morphometrics..."
L111-113: this does require further explanation; why would geometric morphometrics actually allow to capture these differences? It is because it allows to disentangle shape from size variation?
L117: delete “rather than linear morphometrics”
L117: how many species are there in the Artibeus spp. complex? Clarify what portion of the variation is examined here.
L119-120: “which may suggest differences in niche partitioning” should be balanced with a non-adaptive explanation (e.g., “or non-adaptive processes such as drift”). In general, the manuscript strikes me as very adaptationist – not all morphological differences are due to adaptation, as some of the results illustrate.
L122: what is the prediction and justification for examining sexual dimorphism? This should be added.
L128: add justification why this species was selected for the intraspecific study.
L132: underlying causes are discussed, not assessed.

Methods:
L163-165: clarify here and in figure legend -- were the semilandmarks placed evenly spaced along the curve? Add them to the curves in Fig. 2.
L194: means of what?
L196: clarify this is Kmult.
L220: add text to clarify why country was used as a proxy for locality, since this is unusual.
L224: the use of “population” here seems problematic. Replace with “individuals in portions of the geographic range (countries)…”

Results:
L315-318: this could favor non-adaptive hypotheses – a strong phylogenetic signal in size differences and thereby shape via allometry.
L341-342: it’s unclear how these tests were conducted and what is being shown in the table (what are the coefficients shown?). Clarify in methods and Table legend.

Fig 4: change large Artibeus symbols to triangles for readability and consistency.

Discussion:
The discussion section largely recapitulates the results, and is still narrowly focused on phyllostomid bats. For a journal like PeerJ, the results should be discussed in the context of the broader literature on morphological variation, sexual dimorphisms, etc.
L360: replace “successful” with “diverse”.
L381: one possible explanation, if these differences result in performance differences, resource use, etc.
L392: replace “workers” with “studies”.
L416-417: this sentence is very vague; state what you mean more specifically.
L418-419: this sentence suggests that plasticity selects on morphology, but the studies cited do not demonstrate this.
L420-422: this reference is not in the reference list. Did this study actually couple quantitative dietary data with behavioral data and modeling of morphological evolution (or paleontological studies) to demonstrate this?
L433-436: this is assuming the variation leads to functional consequences of importance to access food resources - size may simply override these though.
L444: or other factors that affect body size may “drag” the evolution of skull size and shape with it.
L455-456: unclear what is meant here by “the way size is calculated”
L478-479: how about defense, fighting other males, tent making?
L512-513: there is a rich literature on this for mammals and other vertebrates, expand the scope beyond this narrow group.
L568-569: what is the purpose of this sentence?

---

## Round 0.3 · accepted · Accept

Dear Dr. Hedrick!

I am pleased to accept your study for publication in PeerJ. Excellent work!